# Ngram2vec: Learning Improved Word Representations from Ngram Co-occurrence Statistics

## Abstract

The existing word representation methods mostly limit their information source to word co-occurrence statistics. In this paper, we introduce ngrams into four representation methods: SGNS, GloVe, PPMI matrix and its SVD factorization. Comprehensive experiments are conducted on word analogy and similarity tasks. The results show that improved word representations are learned from ngram co-occurrence statistics. We also demonstrate that the trained ngram representations are useful in many aspects such as finding antonyms and collocations. Besides, a novel approach of building co-occurrence matrix is proposed to alleviate the hardware burden brought by ngrams.

## 1 Introduction

Recently, deep learning approaches have achieved state-of-the-art results on a range of NLP tasks. One of the most fundamental work in this field is word embedding, where low-dimensional word representations are learned from unlabeled corpus. The trained word embeddings reflect semantic and syntactic information of words. They are not only useful in tasks of lexical semantics (e.g. finding similar words), but also widely used as the input of the downstream tasks, such as text classification (Kim, 2014) and tagging (Collobert et al., 2011; Pennington et al., 2014).

Word2vec is one of the most popular word embedding models (Mikolov et al., 2013b,a). It is trained upon <*word, context*> pairs in the local window. Word2vec gains the reputation by its amazing effectiveness and efficiency. It achieves state-of-the-art results on a range of linguistic tasks with only a fraction of time compared with previous techniques. A challenger of word2vec is GloVe (Pennington et al., 2014). Instead of training on <*word, context*> pairs, GloVe directly uses word co-occurrence matrix. They claim that the change brings the improvement over word2vec on both accuracy and speed. Levy and Goldberg (2014b) further reveal that the attractive properties observed in word embeddings are not restricted to neural models. They use traditional bag-of-contexts (concretely, PPMI matrix) to represent words, and achieve comparable results with the above neural embedding models.

The relationships among different representation methods are intricate. A preliminary conclusion is obtained in (Levy et al., 2015), which states that *none of them consistently outperform the other methods.* One should not feel surprised with the conclusion, because these methods all exploit word co-occurrence statistics as the information source and no one goes beyond that. To learn improved word representations, we extend the information source from co-occurrence of *'word-word'* type to co-occurrence of *'ngram-ngram'* type. The idea of using ngrams is well supported by language modeling, one of the oldest problems studied in statistical NLP. In language models, co-occurrence of words and ngrams is used to predict the next word (Kneser and Ney, 1995; Katz, 1987). Actually, the idea of word embedding models roots in language models. They are closely related but with different purposes. Word embedding models aim at learning word representations instead of word prediction. Since ngram is a vital part in language modeling, we are inspired to integrate ngram statistical information into recent word representation methods for better performance.

The idea of using ngram is intuitive. However, there is still rare work using ngrams in recent representation methods. In this paper, we in-

troduce ngrams into SGNS, GloVe, PPMI and its SVD factorization. To evaluate the ngram-based models, comprehensive experiments are conducted on word analogy and similarity tasks. The results demonstrate that the improved word representations are learned from ngram co-occurrence statistics. Besides that, we qualitatively evaluate the trained ngram representations. We show that they are able to reflect ngrams' meanings and syntactic patterns (e.g. 'be + past participle' pattern). The high-quality ngram representations are useful in many ways. For example, ngrams in negative form (e.g. 'not interesting') can be used for finding antonyms (e.g. 'boring').

Finally, a novel method is proposed to build ngram co-occurrence matrix. Our method reduces the disk I/O as much as possible, largely alleviating the costs brought by ngrams. We unify different representation methods in a pipeline. The source code is organized as ***ngram2vec*** toolkit and released at `http://github.com/`.

## 2 Related Work

SGNS, GloVe, PPMI and its SVD factorization are used as baselines. The information used by them does not go beyond word co-occurrence statistics. However, their approaches to using the information are totally different. We review these methods in the following 3 sections. In section 2.4, we revisit the use of ngram in deep learning context.

### 2.1 SGNS

Skip-gram with negative sampling (SGNS) is a model in the word2vec toolkit (Mikolov et al., 2013b,a). Its training procedure follows the majority of neural embedding models (Bengio et al., 2003): (1) *Scan the corpus and use <word, context> pairs in the local window as training samples.* (2) *Train the models to make words useful for predicting contexts (or in reverse).* The details of SGNS is discussed in Section 3.1. Compared to previous neural embedding models, SGNS speeds up the training process, reducing the training time from days or weeks to hours. Also, the trained embeddings possess valuable properties. They are able to reflect relations between the two words accurately, which is evaluated by a fancy task called word analogy.

Due to the above advantages, many models are proposed on the basis of word2vec. For example, Faruqui et al. (2015) introduce knowledge in lexical resources into the word2vec. Zhao et al. (2016) extend the contexts from the local window to the entire texts. Li et al. (2015) use supervised information to guide the training. Dependency parse-tree is used for defining context in (Levy and Goldberg, 2014a). Sub-word information is considered in (Sun et al., 2016; Soricut and Och, 2015).

### 2.2 GloVe

Different from typical neural embedding models which are trained on <*word, context*> pairs, GloVe learns word representation on the basis of co-occurrence matrix (Pennington et al., 2014). GloVe breaks traditional 'words predict contexts' paradigm. Its objective is to reconstruct non-zero values in the matrix. The direct use of matrix is reported to bring more improved results and higher speed. However, There is still dispute about the advantages of GloVe over word2vec (Levy et al., 2015; Schnabel et al., 2015). GloVe and other embedding models are essentially based on word co-occurrence statistics of the corpus. The <*word, context*> pairs and matrix can be converted to each other. Suzuki and Nagata (2015) try to unify GloVe and SGNS in one framework.

### 2.3 PPMI & SVD

When we are satisfied with the embeddings' fancy properties, a natural question is raised: where the properties come from. One conjecture is that it's due to the neural networks. However, Levy and Goldberg (2014c) reveal that SGNS is just factoring PMI matrix implicitly. The experiments also confirm this idea. Levy and Goldberg (2014b) show that positive PMI (PPMI) matrix still rivals the newly proposed embedding models on a range of linguistic tasks. Properties like word analogy are not restricted to neural models. At last, we factorize PPMI matrix via SVD factorization, a classic method for generating low-dimensional vectors from sparse matrix (Deerwester et al., 1990).

### 2.4 Ngram in Deep Learning

In the deep learning literature, ngram has shown to be useful in generating text representations. Recently, convolutional neural networks (CNNs) are reported to perform well on a range of NLP tasks (Blunsom et al., 2014; Hu et al., 2014; Severyn and Moschitti, 2015). CNNs are essentially using ngram information to represent texts. They use 1-D convolutional layers to extract ngram features and the distinct features are selected by max-

pooling layers. In (Li et al., 2016), ngram embedding is introduced into Paragraph Vector model, where text embedding is trained to be useful to predict ngrams in the text. In the word embedding literature, a work that is related to ngram is from (Mikolov et al., 2013b), where phrases are embedded into vectors. It should be noted that phrases are different from ngrams. Phrases have clear semantics and the number of phrases is much less than the number of ngrams. Using phrase embedding has little impact on word embedding's quality.

# 3 Model

In this section, we introduce ngrams into SGNS, GloVe, PPMI and SVD. Section 3.1 reviews the SGNS. Section 3.2 and 3.3 show the details of introducing ngrams into SGNS. In section 3.4, we show the way of using ngrams in GloVe, PPMI and SVD, and propose a novel way of building ngram co-occurrence matrix.

## 3.1 Word Predicts Word: the Revisit of SGNS

First we establish some notations. The raw input is the corpus $T = \{w_1, w_2, ......, w_{|T|}\}$. $W$ and $C$ denote the word and context vocabulary. $\theta$ is the parameters to be optimized. SGNS's parameters involve two parts: word embedding matrix and context embedding matrix. With embedding $\vec{w} \in R^d$, the total number of parameters is $(|W|+|C|)*d$.

The SGNS's objective is to maximize the conditional probabilities of contexts given center words:

$$\sum_{t=1}^{|T|} \left[ \sum_{c \in C(w_t)} \log p(c|w_t; \theta) \right] \quad (1)$$

where $C(w_t) = \{w_i, t - win \leq i \leq t + win \, and \, i \neq t\}$ and *win* denotes the window size. As illustrated in figure 1, the center word 'written' predicts its surrounding words 'Potter', 'is', 'by' and 'J.K.'. In this paper, negative sampling (Mikolov et al., 2013b) is used to approximate the conditional probability:

$$p(c|w) = \sigma(\vec{w}^T \vec{c}) \prod_{j=1}^{k} E_{c_j \sim P_n(C)} \sigma(-\vec{w}^T \vec{c_j}) \quad (2)$$

where $\sigma$ is sigmoid function. $k$ samples (from $c_1$ to $c_k$) are drawn from context distribution raised to the $n$ power.

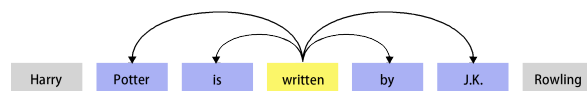

Figure 1: Illustration of 'word predicts word'.

## 3.2 Word Predicts Ngram

In this section, we introduce ngrams into context vocabulary. We treat each ngram as a normal word and give it a unique embedding. During the training, the center word should not only predict its surrounding words, but also predict its surrounding ngrams. As shown in figure 2, center word 'written' predicts the bi-grams in the local window such as 'by J.K.'. The objective of 'word predicts ngram' is similar with original SGNS. The only difference is the definition of the $C(w)$. In ngram case, $C(w)$ is formally defined as follows:

$$C(w_t) = \bigcup_{n=1}^{N} \{w_{i:i+n} | w_{i:i+n} \text{ is not } w_t \text{ AND } \atop t - win \leq i \leq t + win - n + 1\} \quad (3)$$

where $w_{i:i+n}$ denotes the ngram $w_i w_{i+1}...w_{i+n-1}$ and $N$ is the order of context ngram. Two points need to be noticed from the above definition. The first is how to determine the distance between word and ngram. In this paper, we use the distance between word and ngram's far-end word. As show in figure 2, the distance between 'written' and 'Harry Potter' is 3. As a result, 'Harry Potter' is not included in the center word's context. This distance definition ensures that the ngram models don't use the information beyond the pre-specified window, which guarantees the fair comparisons with the baselines. Another point is whether the overlap of word and ngram is allowed or not. In the overlap situation, ngrams are used as context even they contain the center word. As the example in figure 2 shows, ngram 'is written' and 'written by' are predicted by the center word 'written'. In the non-overlap case, these ngrams are excluded. The properties of word embeddings are different when overlap is allowed or not, which is discussed in experiments section.

## 3.3 Ngram Predicts Ngram

We further extend the model to introduce ngrams into center word vocabulary. During the training, center ngrams (including words) predict their

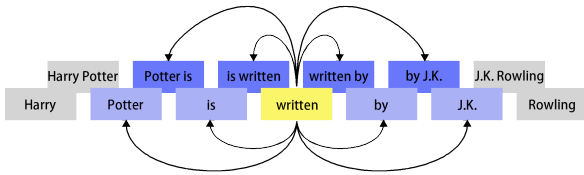

Figure 2: Illustration of 'word predicts ngram'.

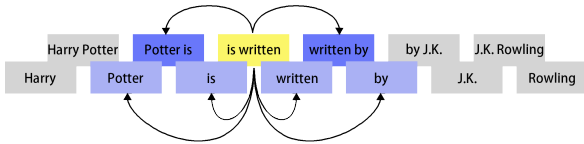

Figure 3: Illustration of 'ngram predicts ngram'.

surrounding ngrams. As shown in figure 3, center bi-gram 'is written' predicts its surrounding words and bi-grams. The objective of 'ngram predicts ngram' is as follows:

$$\sum_{t=1}^{|T|} \sum_{n_w=1}^{N_w} \left[ \sum_{c \in C(w_{t:t+n_w})} log\, p(c|w_{t:t+n_w}; \theta) \right] \quad (4)$$

where $N_w$ is the order of center ngram. The definition of $C(w_{t:t+n_w})$ is as follows:

$$\bigcup_{n_c=1}^{N_c} \{w_{i:i+n_c} | w_{i:i+n_c} \text{ is not } w_{t:t+n_w} \text{ AND} \atop t - win + n_w - 1 \le i \le t + win - n_c + 1\} \quad (5)$$

where $N_c$ is the order of context ngram. To this end, the word embeddings are not only affected by the ngrams in the context, but also indirectly affected by co-occurrence statistics of 'ngram-ngram' type in the corpus.

SGNS is proven to be equivalent with factorizing pointwise mutual information (PMI) matrix (Levy and Goldberg, 2014c). Following their work, we can easily show that models in section 3.2 and 3.3 are implicitly factoring PMI of 'word-ngram' and 'ngram-ngram' type. They are just the representation method discussed in the following section, where ngrams are introduced into positive PMI (PPMI) matrix.

### 3.4 Co-occurrence Matrix Construction

Introducing ngrams into GloVe, PPMI and SVD is straightforward: the only change is to replace word co-occurrence matrices with ngram ones. In the above three sections, we have discussed the

way of taking out <ngram, context> pairs from the corpus. Afterwards, we build co-occurrence matrix upon these pairs. The rest steps are identical with the original baseline models.

| Win | Type | #Pairs |
|-----|------|--------|
| 2 | uni_uni | 0.44B |
| | uni_bi | 0.98B |
| | bi_bi | 2.18B |
| 5 | uni_uni | 1.12B |
| | uni_bi | 2.73B |
| | bi_bi | 6.84B |

Table 1: The number of pairs at different settings. uni_uni, uni_bi and bi_bi respectively denote the models of 'word predicts word', 'word predicts ngram' and 'ngram predicts ngram'. sub-sampling = 1e-5; threshold = 100; overlap.

However, building co-occurrence matrix is not an easy task as it apparently looks like. The introduction of ngrams brings huge burdens on the hardware. The matrix construction cost is closely related to the number of pairs (#Pairs). Table 1 shows the detailed statistics of corpus wiki2010 [1]. We can observe that #Pairs is huge when bi-grams are considered.

To speed up the process of building ngram co-occurrence matrix, we take advantages of 'mixture' strategy (Pennington et al., 2014) and 'stripes' strategy (Dyer et al., 2008; Lin, 2008). The two strategies optimize the process in different aspects. Computational cost is reduced significantly when they are used together.

When the words (or ngrams) are sorted in descending order by frequency, the matrix's top-left corner is dense while the rest part is sparse. Based on this observation, the 'mixture' of two data structures are used for storing matrix. Elements in the top-left corner are stored in a 2D array, which stays in memory. The rest of the elements are stored in the form of <ngram, H> ('stripes' strategy), where H<context, count> is an associative array recording the times the *ngram* and *context* co-occurs. Compared with storing <ngram, context> pairs explicitly, the 'stripes' strategy provides more opportunities to aggregate pairs outside of the top-left corner.

Algorithm 1 shows the way of using the 'mixture' and 'stripes' strategies together. In the first

---

[1] http://nlp.stanford.edu/data/WestburyLab.wikicorp.201004.txt.bz2

stage, pairs are stored in different data structures according to *topLeft* function. Intermediate results are written to temporary files when memory is full. In the second stage, we merge these sorted temporary files to generate co-occurrence matrix. The *getSmallest* function takes out the pair $<ngram, H>$ with the smallest *key* from temporary files. In practice, algorithm 1 is efficient. Instead of using computer clusters (Lin, 2008), we can build the matrix of 'bi_bi' type even in a laptop. It only requires 12GB to store temporary files (win=2, subsampling=0, memory size=4GB), which is much smaller than the implementations in (Pennington et al., 2014; Levy et al., 2015) . More detailed analysis about these strategies can be found in the **ngram2vec** toolkit and the attachment along with our paper [2].

---

**Algorithm 1:** An algorithm for building n-gram co-occurrence matrix

> **Input** : Pairs $P$, Sorted vocabulary $V$
> **Output**: Sorted and aggregated pairs
> 1 The 2D array $A[\,][\,]$;
> 2 The dictionary $D < ngram, H >$;
> 3 The temporary files array $tfs[\,]$; $fid$=1;
> 4 **for** *pair* $p < n, c > in\ P$ **do**
> 5 **if** $topLeft(n, c) == 1$ **then**
> 6 $A[getId(n)][getId(c)]$ += 1;
> 7 **else**
> 8 $D\{n\}\{c\}$ += 1;
> 9 **if** *Memory is full or P is empty* **then**
> 10 Sort $D$ by key (ngram);
> 11 Write $D$ to $tfs[fid]$;
> 12 $fid$ += 1;
> 13 **end**
> 14 **end**
> 15 **end**
> 16 Write $A$ to $tfs[0]$ in the form of $< ngram, H >$;
> 17 $old = getSmallest(tfs)$ ;
> 18 **while** *!(All files in tfs are empty)* **do**
> 19 $new = getSmallest(tfs)$ ;
> 20 **if** $old.ngram == new.ngram$ **then**
> 21 $old =$ $< old.ngram, merge(old.H, new.H) >$;
> 22 **else**
> 23 Write $old$ to disk;
> 24 $old = new$
> 25 **end**
> 26 **end**

---

# 4 Experiments

## 4.1 Datasets

The datasets used in this paper is the same with (Levy et al., 2015), including six similarity and two analogy datasets. In similarity task, a scalar

---

[2]We upload the implementation of algorithm 1 along with the paper

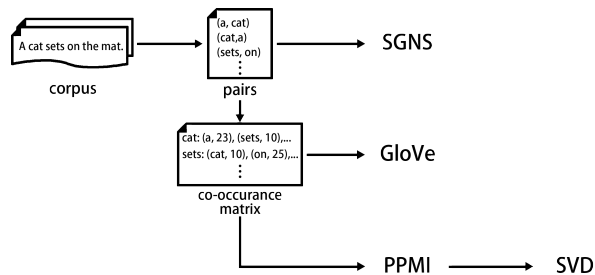

Figure 4: The pipeline.

(e.g. a score from 0 to 10) is used to measure the relation between the two words. The scalar only reflects the strength of the relation, not the type. For example, the 'train, car' pair is given the score of 6.31. In our view, relations are too complex to be reflected by a single score. Suppose that when one is asked to describe the relations between 'Athens' and 'Greece', rather than giving a score, a reasonable answer is probably 'they are both place names and Athens is the capital of Greece'. Besides that, we have to compare pairs with different types in similarity task. Considering the two pairs: 'train, car' and 'movie, star'. It is hard to say which pair has closer relations (higher score) (Schnabel et al., 2015).

In analogy task, relations between the two words are reflected by a vector, which is usually obtained by the difference between word embeddings. Different from scalar, the vector provides more accurate descriptions of relations. For example, capital-country relation is encoded in *vec(Athens)-vec(Greece)*, *vec(Tokyo)-vec(Japan)* and so on. More concretely, the questions in the analogy task are in the form of 'a is to b as c is to d'. 'd' is an unknown word in the test phase. To correctly answer the questions, the models should embed the two relations, *vec(a)-vec(b)* and *vec(c)-vec(d)*, into similar positions in the space. Following (Levy and Goldberg, 2014b), both additive (add) and multiplicative (mul) functions are used for finding word 'd'. The latter one is more suitable for sparse representation.

## 4.2 Pipeline and Hyper-parameter Setting

We implement SGNS, GloVe, PPMI and SVD in a pipeline, allowing the reuse of code and intermediate results. Figure 4 illustrates the overview of the pipeline. Firstly, $<ngram, context>$ pairs are extracted from the corpus as the input of S-GNS. Afterwards, we build the co-occurrence ma-

| Win | Type | | Google Tot. / Sem. / Syn. | | MSR | |
|---|---|---|---|---|---|---|
| | | | Add | Mul | Add | Mul |
| 2 | uni_uni | | .488 / .425 / .540 | .513 / .457 / .559 | .477 | .498 |
| | overlap | uni_bi | .579 / .640 / .528 | .610 / .670 / .560 | .465 | .496 |
| | | bi_bi | .645 / .729 / .576 | .665 / .739 / .603 | .527 | .550 |
| | non-overlap | uni_bi | .589 / .606 / .575 | .610 / .625 / .598 | .469 | .505 |
| | | bi_bi | .651 / .708 / .604 | .682 / .725 / .646 | .525 | .557 |
| 5 | uni_uni | | .587 / .591 / .585 | .607 / .611 / .604 | .445 | .463 |
| | overlap | uni_bi | .635 / .697 / .583 | .638 / .696 / .590 | .448 | .420 |
| | | bi_bi | .649 / .711 / .598 | .660 / .714 / .616 | .463 | .480 |
| | non-overlap | uni_bi | .653 / .666 / .643 | .670 / .676 / .665 | .491 | .516 |
| | | bi_bi | .661 / .687 / .639 | .671 / .690 / .656 | .512 | .536 |

Table 2: Performance of (ngram) SGNS on analogy datasets.

| Win | Type | Sim. | Rel. | Bruni | Radinsky | Luong | Hill |
|---|---|---|---|---|---|---|---|
| 2 | uni_uni | .707 | .565 | .696 | .585 | .450 | .405 |
| | uni_bi | .714 | .568 | .688 | .616 | .471 | .428 |
| | bi_bi | .735 | .594 | .698 | .644 | .468 | .413 |
| 5 | uni_uni | .739 | .637 | .771 | .651 | .457 | .396 |
| | uni_bi | .763 | .624 | .773 | .663 | .471 | .416 |
| | bi_bi | .778 | .647 | .767 | .664 | .473 | .413 |

Table 3: Performance of (ngram) SGNS on similarity datasets.

trix upon the pairs. GloVe and PPMI learn word representations on the basis of co-occurrence matrix. SVD factorizes the PPMI matrix to obtain low-dimensional representation.

Most hyper-parameters come from 'corpus2pairs' part and four representation models. 'corpus2pairs' part determines the source of information for the subsequent models. We basically follow the default setting in (Levy et al., 2015): low-frequency words (ngrams) are removed with a threshold of 100. High-frequency words (ngrams) are removed with sub-sampling at the degree of 1e-5 [3]. Window size is set to 2 and 5. Clean strategy (Levy et al., 2015) is used to ensure no information beyond pre-specified window is included. Non-overlap setting is used in default since it produces much less pairs. For hyper-parameters in the models, we use the embeddings of 300 dimensions. The rest strictly follow the baseline models [4].

We only consider uni-grams (words) and bi-grams in our experiments. Bi-grams are introduced into context (uni_bi) or both context and center 'word' (bi_bi). We suspect that the higher-order grams are so sparse that may deteriorate the performance. The implementation of higher-order

models and their results will be released with ***n-gram2vec*** toolkit.

### 4.3 Ngrams on SGNS

SGNS is a highly popular word embedding model. Even compared with its challengers such as GloVe, SGNS is reported to have more robust performance with faster training speed. Table 2 lists the results on analogy datasets. We can observe that the introduction of bi-grams gives the huge promotions at different hyper-parameter settings. The SGNS of 'bi_bi' type provides the highest results. Bi-grams are very effective for capturing semantic information. Around 20 and 10 percent improvements are witnessed on semantic questions at window sizes of 2 and 5. For syntactic questions, the improvements are still significant. Around 5 percent improvements are achieved on Google (syntactic) and MSR datasets. In traditional language models, ngram is the vital part for predicting next words. Our results confirm the effectiveness of ngrams again on recent word embedding models and more advanced analogy task.

The effect of overlap is large on analogy datasets. Semantic questions prefer the overlap setting. Around 3 percent increase is witnessed compared with non-overlap setting. While in syntactic case, non-overlap setting performs better by a margin of around 5 percent.

Table 3 illustrates the SGNS's performance on similarity task. The use of bi-grams is effective on

---

[3]Sub-sampling is not used in GloVe, which follows its original setting.

[4]http://bitbucket.org/omerlevy/hyperwords for SGNS, PPMI and SVD; http://nlp.stanford.edu/projects/glove/ for GloVe.

| Win | Type | Google | | MSR | | Sim. | Rel. | Bruni | Radinsky | Luong | Hill |
|---|---|---|---|---|---|---|---|---|---|---|---|
| | | Add | Mul | Add | Mul | | | | | | |
| 2 | uni_uni | .488/.571/.418 | .626/.739/.533 | .276 | .441 | .681 | .559 | .670 | .621 | .404 | .366 |
| | uni_bi | .464/.580/.367 | .733/.833/.649 | .286 | .553 | .673 | .532 | .681 | .635 | .401 | .377 |
| 5 | uni_uni | .508/.635/.403 | .588/.757/.447 | .238 | .332 | .732 | .680 | .722 | .626 | .421 | .329 |
| | uni_bi | .509/.682/.365 | .650/.819/.508 | .239 | .416 | .750 | .695 | .708 | .658 | .411 | .371 |

Table 4: Performance of (ngram) PPMI on analogy and similarity datasets.

| Win | Type | Google | | MSR | | Sim. | Rel. | Bruni | Radinsky | Luong | Hill |
|---|---|---|---|---|---|---|---|---|---|---|---|
| | | Add | Mul | Add | Mul | | | | | | |
| 2 | uni_uni | .535/.599/.482 | .540/.610/.481 | .444 | .445 | .681 | .529 | .698 | .608 | .381 | .351 |
| | uni_bi | .543/.601/.493 | .549/.612/.496 | .464 | .472 | .686 | .545 | .695 | .631 | .389 | .352 |
| 5 | uni_uni | .625/.689/.572 | .626/.696/.568 | .476 | .490 | .747 | .600 | .735 | .657 | .389 | .347 |
| | uni_bi | .631/.699/.575 | .633/.703/.574 | .477 | .504 | .752 | .610 | .737 | .631 | .395 | .342 |

Table 5: Performance of (ngram) GloVe on analogy and similarity datasets.

| Win | Type | Google | | MSR | | Sim. | Rel. | Bruni | Radinsky | Luong | Hill |
|---|---|---|---|---|---|---|---|---|---|---|---|
| | | Add | Mul | Add | Mul | | | | | | |
| 2 | uni_uni | .429/.385/.465 | .448/.392/.495 | .345 | .375 | .729 | .611 | .732 | .637 | .499 | .369 |
| | uni_bi | .410/.338/.471 | .446/.364/.514 | .370 | .411 | .731 | .608 | .721 | .637 | .513 | .378 |
| 5 | uni_uni | .436/.415/.454 | .461/.438/.481 | .328 | .332 | .744 | .629 | .756 | .634 | .501 | .337 |
| | uni_bi | .486/.425/.536 | .514/.460/.559 | .349 | .391 | .761 | .642 | .751 | .626 | .522 | .357 |

Table 6: Performance of (ngram) SVD on analogy and similarity datasets.

most cases. However, the improvements brought by bi-grams are not as big as the case in analogy task. The SGNS of 'uni_bi' type outperforms the 'bi_bi' type on most datasets.

### 4.4 Ngrams on GloVe, PPMI, SVD

In this section, we only report the results of models of 'uni_uni' and 'uni_bi' types. Using ngram-ngram co-occurrence statistics bring little improvements but immense costs. Levy and Goldberg (2014b) demonstrate that sparse representation (PPMI matrix) can still achieve competitive results on many linguistic tasks, challenging the dominance of neural embedding models. Table 4 lists the results of PPMI. PPMI prefers Multiplicative (Mul) evalution. To this end, we focus on analyzing the results on Mul columns. When bi-grams are used, significant improvements are witnessed on analogy task. On Google dataset, bi-grams bring around 8 percent increase on the total accuracies. At window size 2, the accuracies even reach 73.3/83.3/64.9. They are pretty high numbers considering only default setting is used. On MSR dataset, around 10 percent improvements are achieved. Though bi-grams are very effective on analogy task, the improvements on similarity task are not witnessed. The PPMI of 'uni_bi' type improves the results of 3 in 6 datasets.

Table 5 and 6 list the GloVe and SVD's results. In GloVe, consistent but minor improvements are achieved on analogy task. Little improvement is witnessed on similarity task. In SVD, bi-grams

sometimes lead to worse results in both analogy and similarity tasks. That's not what we expected before the experiments. We thought ngram would be very effective in GloVe and SVD just like the cases in SGNS, since these models use exactly the same source of information. Our preliminary conjecture is that the default hyper-parameter setting should be blamed. We strictly follow the hyper-parameters used in baseline models, making no adjustments to cater to the introduction of ngrams. Besides that, some common techniques such as dynamic window, decreasing weighting function, dirty sub-sampling are discarded. The relationships between ngrams and various hyper-parameters require further exploration. Though trivial, it may lead to much better results and give researchers better understanding of different representation methods. That will be the focus of our future work.

### 4.5 Qualitative Evaluations of Ngram Embedding

In this section, we analyze the properties of ngram embeddings trained on SGNS of 'bi_bi' type. Ideally, the trained ngram embeddings should reflect ngrams' semantic meanings. For example, the *vec(wasn't able)* should be closed to *vec(unable)*. The *vec(is written)* should be closed to *vec(write)* and *vec(book)*. Also, the trained ngram embeddings should preserve the ngrams' syntactic patterns. For example, 'was written' is in the form of 'be + past participle' and the nearest neighbors

| Pattern | Target | Word | Bi-gram |
|---|---|---|---|
| Negative Form | wasn't able | unable(.745), couldn't(.723), didn't(.680) | was unable(.832), didn't manage(.799), never managed(.786) |
| | don't need | don't(.773), dont(.751), needn't(.715) | dont need(.790), don't have(.785), dont want(.769) |
| | not enough | enough(.708), insufficient(.701), sufficient(.629) | not sufficient(.750), wasn't enough(.729), was insufficient(0.685) |
| Adj. Modifier | heavy rain | torrential(.844), downpours(.780), rain(.766) | torrential rain(.829), heavy rainfall(.799), heavy snow(.799) |
| | strong supporter | supporter(.828), proponent(.733), admirer(.602) | staunch supporter(.870), vocal supporter(.810), a supporter(.808) |
| | high quality | high-quality(.867), quality(.744), inexpensive(.622) | good quality(.813), top quality(.751), superior quality(.736) |
| Passive Voice | was written | written(.793), penned(.675), co-written(.629) | were written(.785), is written(.744), written by(.739) |
| | was sent | sent(.844), dispatched(.661), went(.630) | then sent(.779), later sent(.776), was dispatched(.774) |
| | was pulled | pulled(.730), yanked(.629), limped(.593) | were pulled(.706), pulled from(.691), was ripped(.682) |
| Perfect Tense | has achieved | achieved(.683), achieves(.680), achieving(.625) | has attained(.775), has enjoyed(.741), has gained(.733) |
| | has impacted | interconnectedness(.679), pervade(.676) | have impacted(.838), is affecting(.773), have shaped(.772) |
| | has published | authored(.722), publishes(.705), coauthored(.791) | has authored(.852), has edited(.795), has written(.791) |
| Phrasal Verb | give off | exude(.796), fluoresce(.789), emit(.754) | gave off(.837), giving off(.820), and emit(.816) |
| | make up | comprise(.726), constitute(.616), make(.541) | makes up(.705), making up(.702), comprise the(.672) |
| | picked up | picked(.870), snagged(.544), scooped(.538) | later picked(.712), and picked(.682), then picked(.681) |
| Common Sense | highest mountain | muztagh(.669), prokletije(.664), cadair(.658) | highest peak(.873), tallest mountain(.857), highest summit(.830) |
| | avian influenza | h5n1(.870), zoonotic(.812), adenovirus(.806) | avian flu(.885), the h5n1(.870), flu virus(.868) |
| | computer vision | human-computer(.789), holography(.767) | image processing(.850), object recognition(.818) |

Table 7: Target bi-grams and their nearest neighbours associated with similarity scores.

should possess similar patterns, such as 'is written' and 'was transcribed'.

Table 7 lists the target ngrams and their top nearest neighbours. We divide the target ngrams into six groups according to their patterns. We can observe that the returned words and ngrams are very intuitive. As might be expected, synonyms of the target ngrams are returned in top positions (e.g. 'give off' and 'emit'; 'heavy rain' and 'downpours'). Besides that, from the results of the first group, it can be observed that bi-gram in negative form 'not X' is useful for finding the antonym of word 'X'. The trained n-gram embeddings also preserve some common sense. For example, the returned results of 'highest mountain' is a list of mountain names (with a few exceptions such as 'unclimbed'). In terms of syntactic patterns, we can observe that in most cases, the returned ngrams are in the similar form of target ngrams. In general, the trained embeddings basically reflect the semantic meanings and syntactic patterns of ngrams.

With high-quality ngram embeddings, we have the opportunity to do more interesting things in our future work. For example, we will construct a antonym dataset to evaluate ngram embeddings systematically. Besides that, we will find more scenarios for using ngram embeddings. In our view, ngram embeddings may be useful in many NLP tasks. For example, Johnson and Zhang (2015) use one-hot ngram representation as the input of CNN. Li et al. (2016) use the ngram embeddings to represent texts. Intuitively, initializing these models with pre-trained ngram embeddings may further improve the accuracies.

## 5 Conclusion

We introduce ngrams into four representation methods. The experimental results demonstrate ngrams' effectiveness for learning improved word representations. In addition, we find that the trained ngram embeddings are able to reflect their semantic meanings and syntactic patterns. To alleviate the costs brought by ngrams, we propose a novel way of building co-occurrence matrix, enabling the ngram-based models to run on cheap hardware.

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
