# Peer review of "Ngram2vec: Learning Improved Word Representations from Ngram Co-occurrence Statistics"

_ACL 2017 — decision unknown_

[Official Review · Reviewer 1 · rating 4 · confidence 4]
soundness 5 · originality 3 · clarity 5 · impact 3 · substance 4 · appropriateness 5 · meaningful comparison 5 · presentation format Oral Presentation

- Strengths:
This paper presents an extension of many popular methods for learning vector
representations of text.  The original methods, such as skip-gram with negative
sampling, Glove, or other PMI based approaches currently use word cooccurrence
statistics, but all of those approaches could be extended to n-gram based
statistics.  N-gram based statistics would increase the complexity of every
algorithm because both the vocabulary of the embeddings and the context space
would be many times larger.  This paper presents a method to learn embeddings
for ngrams with ngram context, and efficiently computes these embeddings.  On
similarity and analogy tasks, they present strong results.

- Weaknesses:
I would have loved to see some experiments on real tasks where these embeddings
are used as input beyond the experiments presented in the paper.  That would
have made the paper far stronger.

- General Discussion:
Even with the aforementioned weakness, I think this is a nice paper to have at
ACL.

I have read the author response.

[Official Review · Reviewer 2 · rating 4 · confidence 2]
soundness 5 · originality 3 · clarity 4 · impact 3 · substance 4 · appropriateness 5 · meaningful comparison 5 · presentation format Oral Presentation

- Strengths: The idea to train word2vec-type models with ngrams (here
specifically: bigrams) instead of words is excellent. The range of experimental
settings (four word2vec-type algorithms, several word/bigram conditions) covers
quite a bit of ground. The qualitative inspection of the bigram embeddings is
interesting and shows the potential of this type of model for multi-word
expressions. 

- Weaknesses: This paper would benefit from a check by a native speaker of
English, especially regarding the use of articles. The description of the
similarity and analogy tasks comes at a strange place in the paper (4.1
Datasets). 

- General Discussion: As is done at some point well into the paper, it could be
clarified from the start that this is simply a generalization of the original
word2vec idea, redefining the word as an ngram (unigram) and then also using
bigrams. It would be good to give a rationale why larger ngrams have not been
used.

(I have read the author response.)

[Official Review · Reviewer 3 · rating 3 · confidence 4]
soundness 5 · originality 3 · clarity 3 · impact 3 · substance 3 · appropriateness 5 · meaningful comparison 5 · presentation format Poster

This paper modifies existing word embedding algorithms (GloVe, Skip Gram, PPMI,
SVD) to include ngram-ngram cooccurance statistics. To deal with the large
computational costs of storing such expensive matrices, the authors propose an
algorithm that uses two different strategies to collect counts.  

- Strengths:

* The proposed work seems like a natural extension of existing work on learning
word embeddings. By integrating bigram information, one can expect to capture
richer syntactic and semantic information.

- Weaknesses:

* While the authors propose learning embeddings for bigrams (bi_bi case), they
actually do not evaluate the embeddings for the learned bigrams except for the
qualitative evaluation in Table 7. A more quantitative evaluation on
paraphrasing or other related tasks that can include bigram representations
could have been a good contribution.

* The evaluation and the results are not very convincing - the results do not
show consistent trends, and some of the improvements are not necessarily
statistically significant.

* The paper reads clunkily due to significant grammar and spelling errors,
and needs a major editing pass.

- General Discussion:

This paper is an extension of standard embedding learning techniques to include
information from bigram-bigram coocurance. While the work is interesting and a
natural extension of existing work, the evaluation and methods leaves some open
questions. Apart from the ones mentioned in the weaknesses, some minor
questions for the authors :

* Why is there significant difference between the overlap and non-overlap
cases? I would be more interested in finding out more than the quantitative
difference shown on the tasks.

I have read the author response. I look forward to seeing the revised version
of the paper.